# Retrospect and prospect of a section-based stratigraphic and palaeontological database -- Geobiodiversity Database

Hong-He Xu[1,3, *], Zhi-Bin Niu[1,2, *] Yan-Sen Chen[1,3]

[1] State Key Laboratory of Palaeobiology and Stratigraphy, Nanjing Institute of Geology and Palaeontology, Chinese

5  Academy of Sciences, 210008 Nanjing, China

2 College of Intelligence and Computing, Tianjin University, 300354 Tianjin, China

Center for Excellence in Life and Paleoenvironment, Chinese Academy of Sciences, 210008 Nanjing, China

* The authors contributed equally to this work.

*Correspondence to*: Hong-He Xu (hhxu@nigps.ac.cn) and Zhi-Bin Niu (zniu@tju.edu.cn)

**Abstract**

Big data are significant to the quantitative analysis and contribute to the data-driven scientific research and discoveries. Here the thorough introduction is given on the Geobiodiversity database (GBDB), a comprehensive stratigraphic and palaeontological database. The GBDB includes abundant geological records from China and contributes a serial of scientific studies on early Palaeozoic palaeogeography, tectonic and biodiversity evolution of China. Nevertheless, the existing problems

of the GBDB limited the using of its data. The turnover and improvement of the GBDB were started in 2019. Besides the data collecting, processing and visualization as the GBDB did previously, the database and the website are optimized and re-designed, the new GBDB working team pays more attention to data analyzing with the professional artificial intelligence techniques. GBDB is complementary to other related databases, and further collaborations are proposed to mutually benefit and push forward the quantitative research of palaeontology and stratigraphy in the era of big data. The datasets (Xu, 2020)

are freely downloadable from http://doi.org/10.5281/zenodo.3667645.

**Introduction**

Palaeontology and stratigraphy have become a quantitative study as a branch discipline of geoscience and there has been a subsequent rapid increase in the implementation of numerical methods in palaeontology (Hammer and Harper, 2006) and stratigraphy (Kemple et al., 1995; Rong et al., 2007; Huang et al., 2012; Fan et al., 2013b). Quantitative analysis based on big

data of fossil and stratum records have been more common recently, especially on biodiversity evolution (Alroy et al., 2008; Hautmann, 2016; Fan et al., 2020), graphic correlation of strata (Fan et al., 2013b), palaeoecology (Muscente et al., 2018), mass extinction (Muscente et al., 2019) and palaeogeography (Hou et al., 2020). There are professional databases, such as Paleobiology Database (PBDB), Macrostrat (https://macrostrat.org/) and Geobiodiversity Database (GBDB), storing and providing a big volume of fossil record data and making these high-impact studies possible. Well-structured stratigraphic and

palaeontological databases and friendly-accessible data are significant to the quantitative development of the discipline and furthermore, push forward the digital Earth science in the era of big data (Guo, 2017). In this paper, we introduce a comprehensive database of stratigraphy and palaeontology biodiversity, Geobiodiversity Database (GBDB), being registered as the intellectual property of the Nanjing Institute of Geology and Palaeontology (NIGP), Chinese Academy of Sciences (CAS), its brief history, development, and improvement. The comparisons between related databases are also given.

**2. A brief history of the Geobiodiversity Database**

The Geobiodiversity Database (GBDB) was started in 2006 and provided online service since 2007 when there was a strong and urgent demand for the quantitative understanding of fossil and stratum records from China (Rong et al., 2006; 2007). At that time there had been a large paleobiology database that included plenty of fossil occurrence data from the publications of euro-languages, however, fossil and stratum data from China were temporarily ignored for the obstacle of language. The

purpose of the GBDB establishment was to accommodate fossil and stratum data, geological sections as well as fossil collections, especially from Chinese publications.

Since the start of the GBDB, there used to be at most ten data entry clerks, including master or PhD students, assistant researchers and non-professional employees, digitalizing palaeotological and stratigraphic descriptions "from the page into cyberspace" (Normile, 2019) and aligning these data with the same standards that is acceptable by international researchers,

so that a researcher could quickly link them to carry on quantitative analysis that used to be rare previously.

The GBDB was designed to facilitate regional and global scientific collaborations focused on palaeobiodiversity, systematics, palaeogeography, palaeoecology, regional correlation, and quantitative stratigraphy.

The basic functions of the data input and output were gradually provided. In 2013, a serial type of palaeontological and stratigraphic data were included in the GBDB, such as taxonomy, identification features, occurrence, opinion, lithostratigraphy,

biostratigraphy, chemostratigraphy, radio isotopic dating, reference and palaeogeographic map (Fan et al., 2013a; Fan et al., 2014). Additionally, there were embedding a few online statistic and visualization tools, such as Time Scale Creator (integrated into GBDB in 2010), a stratigraphic visualization tool designed by Jim Ogg and Adam Lugowski (http://www.tscreator.com), and 2) GeoVisual (integrated in GBDB in 2010 and updated in 2012), a tool used for geographic visualization and preliminary biogeographic analysis.

One of the exclusive features of the GBDB is its abundant geological section data, which is readily exported for the software of strata correlation, such as Constrained Optimization (CONOP) (Kemple et al., 1995) and SinoCor. The SinoCor was designed and updated by Fan et al. (2002) and Fan and Zhang (2000; 2004). Its correlation resembles the CONOP but requires a unique file format, resulting in few users except the developers themselves (Fan et al., 2013b). The SinoCor 4.0 was declared as the latest version but the formal publication and further developing is still awaiting (see Fan et al., 2013a; b).

The GBDB became the formal database of the International Commission on Stratigraphy in August 2012 at the 34[th] International Geological Congress in Brisbane, Australia, and, as a result, a new major goal of the GBDB is to integrate stratigraphic standards (e.g. the GSSPs) with comprehensive and authoritative web-based stratigraphic information service for global geoscientists, educators and the public.

     Since 2011, data related to early Paleozoic, especially Ordovician and Silurian periods, stratigraphic and palaeontological

records had been quantitatively analyzed and a serial of scientific findings were achieved. The related research themes include the Ordovician and Silurian palaeogeography and tectonic evolution of South China (Chen et al., 2012; 2014b; 2017a), the spatio-temporal pattern of the Ordovician and Silurian marine organisms from China (Chen et al., 2014a; 2017b; Zhang et al., 2014a; 2016), and the Paleozoic paleogeography evolution of South China (Chen et al., 2018; Zhang et al., 2014b; Hou et al., 2020). Recently, nearly all data of Paleozoic marine organisms of GBDB were used to analyze the biodiversity evolution (Fan

et al., 2020). Though all data were from China, the Paleozoic geological sections of China actually covered several palaeocontinents and might reflect the global biodiversity change.

     In 2017, the GBDB became a data partner of the British Geological Survey (BGS) and started to digitalize the fossil and stratum data and establish the datasets for the BGS. This is a time-taking job and still is carrying by the GBDB data entry team, for the BGS amassed and housed about 3 million fossils gathered over more than 150 years at thousands of sites across the

British Islands.

     At the end of 2018, the head of the GBDB, Dr. Fan J.X., left the NIGP, CAS and Dr. Xu H.H. took over the GBDB. Besides the data collecting, processing and visualization as the GBDB group did during 2007-2018, data of fossil terrestrial organisms, such as insects and plants, were input into the GBDB, the database and the website were optimized and re-designed, and furthermore, the new GBDB working team pays more attention to the data analyzing, a professional artificial intelligence

working group is joining for data analyzing. The GBDB is ushering a new start.

### 3. The data of the Geobiodiversity Database

     The Geobiodiversity database (GBDB) was designed as a stratigraphic and palaeontological database and its input format was designed as geological section-based, which means that data entry clerks or any scientific users must input the metadata for the GBDB according to the geological sections or virtual sections. Every metadata record contains all geological

information of a geological section, including its basic unit (or bed or layer), sediment color, lithology, thickness, horizon, locality, palaeo-block, geological age, bio-stratigraphy, geochemistry, palaeo-ecology, radio isotopic age, fossil collection and any available original information of the rock specimens or fossil sample during the fieldwork. An individual geological section normally can be subdivided into dozens of basic units when it is input the GBDB. Such geological section records with much information can be readily found from stratigraphic and palaeontological literature. However, many paleontological



descriptions or reports are lacking detailed stratigraphic description, the GBDB includes these records as virtual sections,
       which has only a very portion, for example, of a single bed or collection of the whole section. The borehole core record is also
       input into the GBDB as a virtual section (Figure 1).

       The stratigraphic data in GBDB were based on those published in Chinese literature since the 1920s. By November 2019,
       all stratigraphic horizons and nearly all published geological sections can be browsed in the GBDB (Figures 2, 3). It is
noteworthy that in the GBDB there isn't direct record of fossil occurrences, which, however, are related to the stratigraphic
       records of the database and included in the palaeontological data of the GBDB. The palaeontological data are from the fossil
       collection of individual geological sections and borehole cores, including taxonomy (species, genus, family, order, class and
       division), major group, synonym (opinion data) and description (key features) (Figure 1). Though the GBDB is geological
       section-based, from which fossil occurrences can be output, it is compatible with fossil occurrence-based database. Most fossil
collections and occurrences of all sections from China were included in the GBDB (Figure 3). Subsequent authors in further
       study amended a portion of fossil taxa from these sections. In this way, there are also a plenty of opinion data in the GBDB.

       Since 2017, the GBDB started to record the data of Global Boundary Stratotype Section and Point (GSSP) of the
       International Commission on Stratigraphy, including the detail information of GSSP and some panorama and three-
       dimensional scanning of individual GSSP.

Since August 2017, British Geological Survey (BGS) and GBDB started to collaborate in stratigraphic and palaeontological
       data processing. GBDB data working team help to digitalize the geological reports from the BGS archive and to build isolated
       datasets for it. This part of work is still ongoing and much work of data sorting, cleaning and checking are still awaiting.

       Since 2019, the GBDB starts to include the borehole core data of petroleum companies, such as China National Offshore
       Oil Corporation and China National Petroleum Corporation.

In brief, as much as possible stratigraphic and palaeontological records are collected from the original geological
       publications. Since the establishment, the GBDB data team consciously collected and included stratigraphic and
       palaeontological data from Chinese literature. The detailed statistic outcome is given here for the first time (Table 1) (see Xu,
       2020).

       **Newly-added data in the GBDB**

For a long time, the biodiversity evolution study was based on marine organism fossil records. For example, the earliest
       quantitative analysis of the geological time biodiversity that draws the conclusion of the five mass-extinction (Raup and
       Sepkoski, 1982) and a serial of related geological biodiversity studies were based on marine organism fossil family or genus
       records (Jablonski, 1994; Rong et al., 2006; 2007; Alroy et al., 2008).

       The quantitative study based on terrestrial organism fossil records is relatively less in spite of there has been a number of
palaeontological studies of terrestrial organisms. There used to be quantitative studies on the plant diversity of the Silurian and



Devonian periods that was significant for the early plant evolution and diversification (Xiong et al., 2013) and the study on plant diversity change during the Permian-Triassic boundary (Xiong and Wang, 2007) that is the very time of the greatest mass extinction of the geological history wiping off over 95% marine organisms (Jablonski, 1994). Both plant diversity studies used fossil record data from South China and listed the data as the supplementary material of the published papers. It took the authors of the two studies a few years to complete the data collection, even the data were only from South China palaeo-block.

An inconvenient fact is that the database of fossil terrestrial organisms is not as good as that of marine ones. Based on this, it took the GBDB over a year to complete the related database and to collect related data consciously. GBDB now has exclusive databases for the fossil terrestrial organisms. By far, the fossil plant record dataset has collected 738 Devonian plant species occurrences from global localities and thousands of Mesozoic plant species occurrences from China. These data will be included into the GBDB after the work of data formatting and cleaning.

**Fossil insect records in the GBDB**

The insect fossil records in the GBDB were little because few accurate insect fossil occurrences or collections were recorded with geological section descriptions. Additionally, plenty of insect fossils were found from ambers instead of lithological horizons. As a result, a number of fossil insect studies were carried directly without detailed stratigraphic descriptions. Fossil insect occurrences and collections are not closely related to their lithological horizons. The insect fossil records in the GBDB greatly increase after taking over the international fossil insect database of the International Palaeoentomological Society, EDNA (https://fossilinsectdatabase.co.uk/), which holds details of the holotypes of all fossil insects in the world.

The EDNA was named after Edna Clifford who started the recording of new species on a card index system and was designed as an update of Handlirsch's 1906-1908 "Die Fossilen insekten und die phylogenie der rezenten formen" which listed all the then known fossil insect species. Handlirsch recorded 5 160 species in 1906. The database is detailed in its contents: it records taxonomic information, synonym details, references for every species (including the page number where it is introduced), and for holotypes site details, stratigraphic information, and geological details are recorded. All the data has been obtained from exhaustive literature searches.

The EDNA aims to be a complete, fully interactive, list of all the species of insects named from the fossil record, with the site, geological age, and reference for each holotype. Updating and checking will be ongoing, and the data available will be greatly improved if details of omissions and errors are sent to the administrator for incorporation. The data comes from an exhaustive literature search and in the 2019 edition contains 28 439 species names (including synonyms) extracted from 5218 references (Figure 3d). The data is held in 38 fields, all of which are searchable, independently or in combination, and the output can contain any one or more as required.

Fields include: generic and specific names, citation, subfamily, family, superfamily, division, suborder and order: Author, title, journal, and date of publication, and page on which the species is first described: Time data including stage, epoch,

subperiod, period and era and age (range) in millions of years: Bed, member, formation, and group: Site name, nearest feature (town, river etc.) county, state, country and continent (Figure 4). For all taxonomic ranks, citations can be included and both junior and senior synonyms displayed. Natural History Museum London Library call numbers are also included.

**4. Database comparisons and discussions**

The section-based Geobiodiversity database is different from the fossil occurrence-based Paleobiology Database (PBDB), which was founded in 1998 and became the largest paleobiological database. Data includes fossil taxa, collection, opinions (paleobiological views from different authors) and even related publications. The data volume of the PBDB is larger than the GBDB (Table 1). The noticeable difference lies in that the PBDB has little information about geological sections. Whilst the GBDB is known as its large number of geological sections.

By November 2019, 26 450 geological sections are recorded in the GBDB, the geological age of which ranging from Ediacaran to Eocene. These include nearly all sections and a part of borehole cores from China, worldwide sections and borehole cores from open publications and reports of the British Geology Survey. Every record is based on published literature or internal reports. This explains that GBDB has more references than the PBDB (Table 1).

As we mentioned, the GBDB is geological section-based; every record was subdivided into detailed parts when being input in the database. The fossil occurrence and collection data can also be exported from the GBDB, just as those in the PBDB. Nevertheless, the fossil taxon number recorded in GBDB is about 30% of that in PBDB, whilst the fossil occurrence records in GBDB is about 40% of that in PBDB (Table 1). This is because the two databases have different histories, the PBDB was founded in 1998, the GBDB, in 2007. The second reason for the difference in data quantity of the two databases is that for a long time the GBDB had focused on stratigraphic records instead of only fossils, and the palaeontological information had been input as complementary items of individual stratigraphic data (Figure 1).

The stratigraphic study and recording in the GBDB are reminiscent of Macrostrat (https://macrostrat.org/), which is a platform for the aggregation and distribution of geological data relevant to the spatial and temporal distribution of sedimentary, igneous, and metamorphic rocks as well as data extracted from them. Macrostrat aims to become a community resource for the addition, editing, and distribution of new stratigraphic, lithologic, environmental, and economic data. By November 2019, Macrostrat records 1 534 regional rock columns, 35 163 rock units, and 2 484 619 geologic map polygons. It is also worth noting that Macrostrat records most geological data from North America, whilst the GBDB includes nearly all stratigraphic data from China.



### 5. Problems, improvements and prospects

The website of GBDB was started online in 2007, few updating was given since that time. According to the feedback received from the users of the GBDB, the existing problems of the website are listed as followings.

1) The website is developed using Net Framework 2.0, which is out of date and results in that the interface and layout of the website are not readily to update, and furthermore, the website would lose pages or has no response when querying.

2) The data volume of the individual datasets is neither visible nor searchable.

3) The data query is not friendly, only the geographic and horizon terms can be used as keywords for searching.

4) Data is not readily accessible. Only the registered users have access to the data, but the new registration requires the activation of the web administrator.

5) Data download is not convenience or friendly. The downloading process includes several steps of selecting the data to the extra dataset and exporting the data from the dataset.

6) The data format is not well compatible with that of other databases.

Additionally, the backup mechanism is not using in the GBDB and its data is potentially hazardous. Updating and improvement to the GBDB and its website are necessary to make the data widely used.

We comprehensively updated the server and the website of the GBDB, making the database a safe data bank and the website a new and friendly portal. The new website has the optimized input and output of data, the search engine, and the data examination system.

After the first step of inputting, the raw data will be checked by registered authorizers, such action aims to make sure that the data conform to the publication but not to the authorizer's point of view. Only the checked records go into the database. No matter the identity of the enterer, the procession of the data input is in such way.

In the previous version of the GBDB website, registered users who would like to download data need to search and select certain lithologic units and build a temporary dataset. Only the data in this dataset can be downloaded unfriendly. The new website simplifies this process, no temporary dataset is required. Any user can search and download interesting data directly. Additionally, the types of output data are compatible with occurrence-based data. The user of GBDB can obtain both section-based stratigraphic and fossil occurrence data.

Data export format includes the regular spreadsheet, such as Excel, CSV, and computer-readable JSON files. An exclusive spreadsheet form is designed for the geological sections in the GBDB. Its structure is better matching the geological column and can be output into the graph readily.

The updates and new features of the GBDB 2.0β also include:

1) Data visualization is developed. All data are plotted on the world map of the homepage that also displays the volume of the all data in the right up corner. The view center is the map of China and the map can be zoomed in or out using mouse scroll.

Geological sections are showed as individual spots and their rough or detailed information can be checked easily.

2) The different colors of spots on the map correspond to various geological stages of the International Chronostratigraphic Chart that is show in disk-shaped in the right lower corner and can be hidden manually.

3) Interactive experience is enhanced on a friendly interface. The downloading data is simplified and optimized and. Any visitor can download data but only the registered users can add data. The data examination is developed.

4) The data retrieval and query are optimized. One can search and download from the homepage.

5) The case study and publications based on GBDB data are listed in the 'Research' button, which also shows the people and funds concerning the GBDB.

6) The palaeogeographic map layer is added, all data can be plotted on it.

7) User system is optimized; personal profile and favorite can be customized set.

8) The old version of the GBDB is remained and has an entrance in the homepage.

In the next step, more data visualization and analytic tools (Figure 5) will be embedded in the new GBDB website publicly, 225 for stratigraphic and palaeontological researching with the advance of network analysis, machine learning, and cognitive visual analysis.

It is a consensus that all scientific data belong to the global scientific community. Everyone has free access to the data recorded in the three databases, PBDB, GBDB, and Macrostrat, all data are freely used for quantitative analysis and serving scientific researching but not commercial purposes.

GBDB and PBDB are complementary in the quantitative palaeontological study. The two databases include thorough records of fossil occurrences. Fossil taxa of the two databases contain not only the widely-distributed and endemic fossils; and those published in both English (and others) and Chinese languages. GBDB and Macrostrat are complementary in the stratigraphic study to some extent. The data of the two databases contain records from both North America and China. Data from these databases, therefore, provides the possibility to conduct various stratigraphic and paleontological analyses. Potential 235 and comprehensive knowledge is hidden in these data.

**Author contribution:** HX and ZN equally designed the project, developed the model, and performed the simulations. HX prepared the manuscript with contributions from ZN. Y-SC gave technician supports.

**Competing interests:** The authors declare that they have no conflict of interest.

**Data availability:** All data are downloadable from the website portal https://www.geobiodiversity.com/ or http://doi.org/10.5281/zenodo.3667645 (Xu, 2020).





**Acknowledgments**

We thank Prof. W. Kiessling, Friedrich-Alexander-Universität Erlangen-Nürnberg, Germany; Drs. Na Lin, Li Qijian and Wang Bo, Nanjing Institute of Geology and Palaeontology, Chinese Academy of Sciences (CAS); Dr. Pan Zhaohui, Institute of

Vertebrate Paleontology and Paleoanthropology, CAS, and Mr. Wu Junqi, College of Intelligence and Computing, Tianjin University, for constructive suggestions and helps. This research was supported by the Strategic Priority Research Program of the Chinese Academy of Sciences (Grants XDA19050101 and XDB26000000) and National Natural Science Foundation of China (Grants 41772012 and 61802278).

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



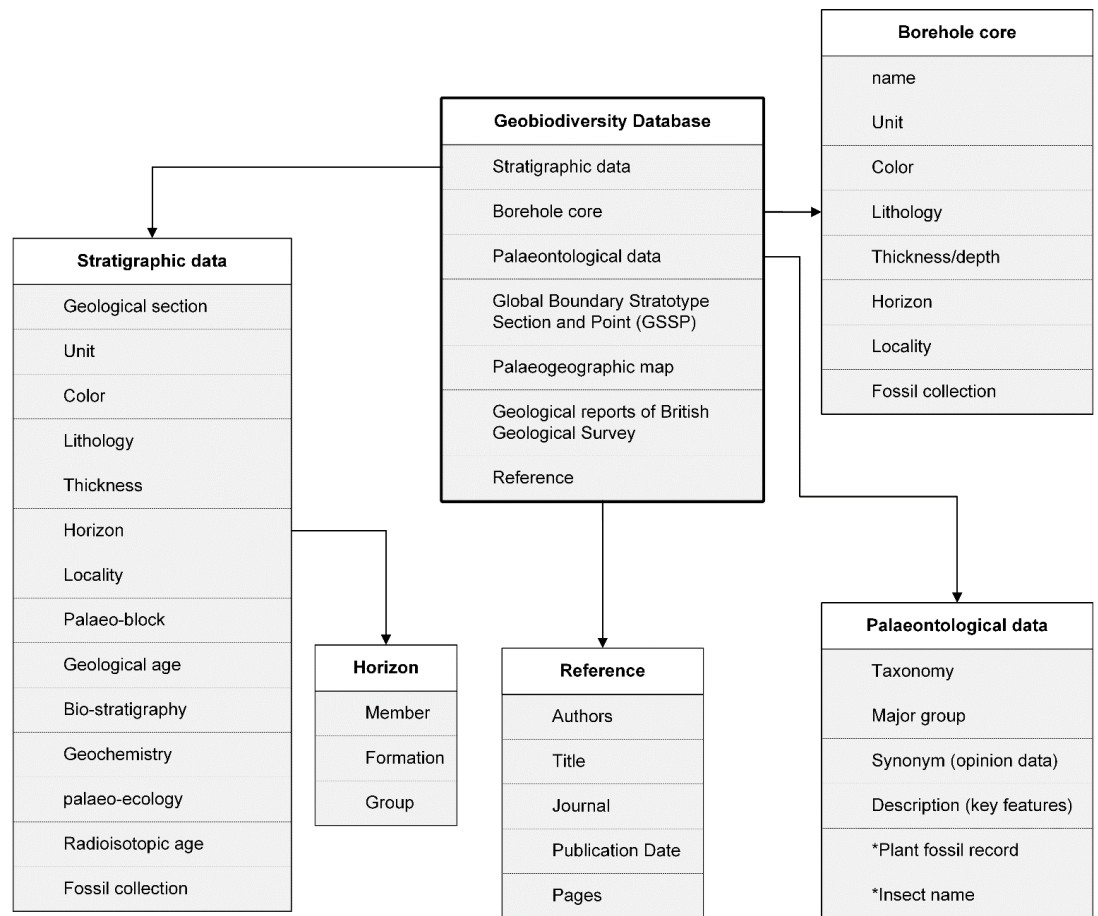


**Figure 1. The data structure of the Geobiodiversity Database (GBDB). * refers the newly-added datasets.**



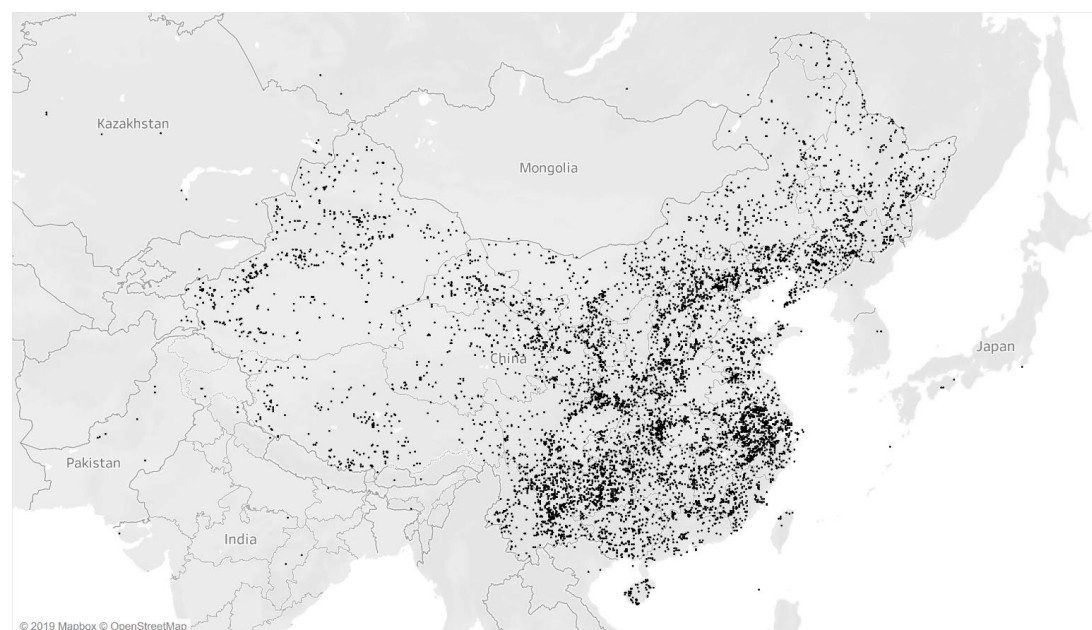

**Figure 2. Regional (China-East Asia) distribution of stratigraphic and palaeontological data (2007-2018) of the Geobiodiversity Database (GBDB) (Xu, 2020). Every black dot corresponds a stratigraphic or palaeontological record of the GBDB. The map: © OpenStreetMap contributors 2020. Distributed under a Creative Commons BY-SA License.**


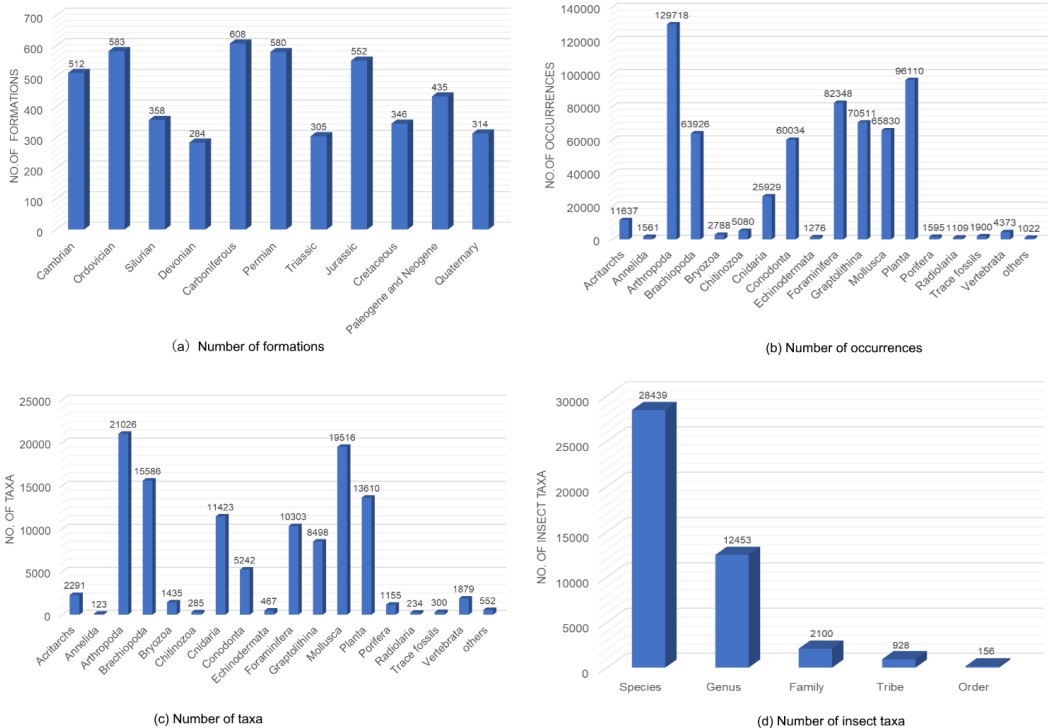

**Figure 3. Histograms showing the statistic outcome of the data in Geobiodiversity Database (GBDB), the detailed numbers are shown**





on every item. (a) Stratigraphic formations of different ages from China. (b, c) Fossil taxa and occurrences of different groups. (d) Newly-added taxa of the Class Insecta, these taxa are not included in the statistic outcome of the Table 1.

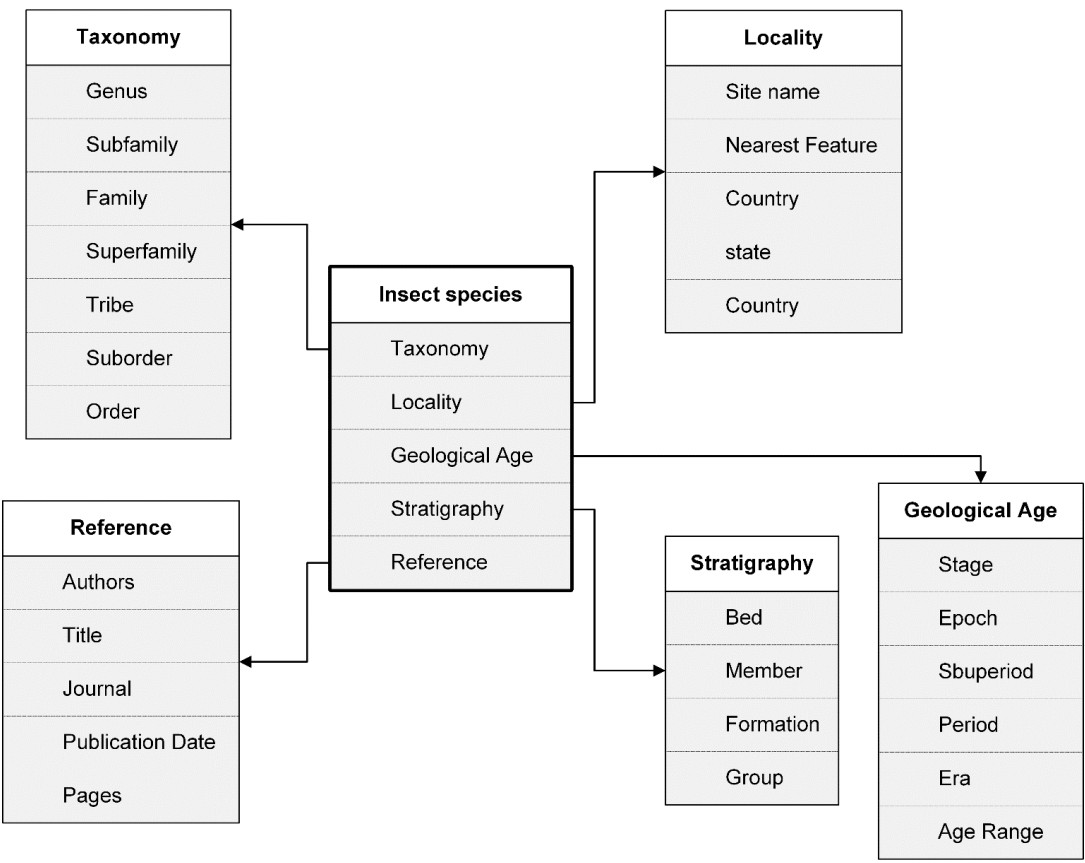

**Figure 4. The data structure of insect species name dataset of the Geobiodiversity Database (GBDB) (Xu, 2020).**

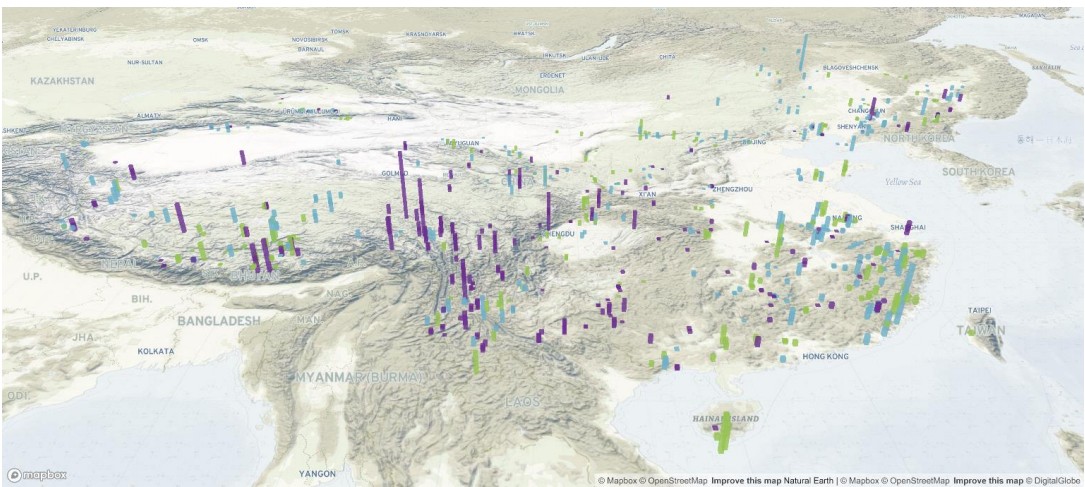

**Figure 5. The screenshot of a three-dimensional bar graph visualizing the Mesozoic stratigraphic formations from China. Data are from the Geobiodiversity Database (GBDB). The colors of the bars are based on those in the International Chronostratigraphic**



**Chart: Triassic (pink), Jurassic (blue) and Cretaceous (light green). The map: ©OpenStreetMap contributors 2020. Distributed under a Creative Commons BY-SA License.** The website of this graph is: http://167.71.205.3/3dbar/

Table 1. The comparison of the two widely-used palaeontological databases. Note that the newly-added data of terrestrial organisms, plant and insect fossil records, are not included in the GBDB statistic outcome (by November 2019).

|  | Paleobiology Database (PBDB) | Geobiodiversity Database (GBDB) |
|---|---|---|
| Type | fossil occurrence-based | section-based |
| No. of references | 69 248 | 96 511 |
| No. of taxa | 388 533 | 113 925 |
| No. of opinions | 718 165 | 18 058 |
| No. of collections | 202 189 | 124 456 |
| No. of occurrences | 1 414 981 | 626 747 |
| No. of sections | n/a | 26 423 |
| No. of formations | 16 252* | 4 736 |
| No. of publications | 344 | 45 |
| Founded at | 1998 | 2007 |
| Website | https://paleobiodb.org/ | http://geobiodiversity.com/ |

*The stratigraphic formation data of the PBDB was obtained from Prof. W. Kiessling whilst one can see these records from the
portal of the PBDB.