# Peer review of "A status report on a section-based stratigraphic and palaeontological database – the Geobiodiversity Database"

_Earth System Science Data, 2020_

## Referee Comment (RC1) · Peter Sadler (Referee) · 11 Aug 2020

GENERAL COMMENTS ON THE TEXT: It is good to learn that the GBDB (Geobiodiversity Database) may be fully back-in-business. As a section-based database, it had enjoyed a useful position among taxon-based and collection-based information hubs. It has been the foundation of groundbreaking publications in macroevolution and a test bed for correlation tools. Speculation about its fate had begun when access became more challenging and updates ceased. There followed stories to the effect that the trusted original team had quit for untold reasons and there were rumors of a law suit. My comments are intended to improve the positive impact of this article on potential

users of the resuscitated GBDB.

Some research paleobiologists and stratigraphers trust public databases only as a way to find the primary literature. They like to do their own quality control and are hard to convince that a database management team can be as thorough as themselves. Database managers must be very careful not to undermine the credibility of their database in the minds of potential users. User-trust is a fragile commodity, easily lost. This retrospective article risks undermining the credibility of the GBDB in some ways, especially with its limited historical overview and veiled criticism of the former management. A full update summary that describes the details of new and tested functionality would do more to restore the trust of professional users.

Retrospection is a strange approach to an update. Effective product updates look forward, optimistically, not backward; they should not waste time criticizing prior functionality. Given the speculations about the change of management of the GBDB, dwelling on the criticisms could seem more political than intellectual. Previous user criticisms can be presented positively as wishes fulfilled.

Retrospection is, at best, a chance to display depth of knowledge about the longer history of numerical stratigraphy and the databases that came to support it. Here the article falls short, especially in the introduction. It is widely known that the GBDB came late to the suite of community databases that was already led in paleontology by the PBDB (PaleoBiology DataBase). Insightfully advised, the GBDB made a new niche for itself – a section-based structure and the promise of access to Chinese locality data. The longer-established PBDB was founded on the database compiled by Jack Sepkoski, a member of the Chicago school, under Dave Raup, that had pioneered numerical paleobiology. The same school trained the founder of the Macrostrat database. The introduction does not properly acknowledge this deep history and offers nothing on the "Fossil Record" database, a long-standing collaboration of museums, universities and government agencies in New Zealand. Nor is Neptune mentioned; that database was built for the Deep Sea Drilling Project and its successors. Other geosciences, notably seismology, were substantially ahead in pioneering and funding big community databases, but even here the separately funded organizations have recently shown some fragility.

The introduction also attempts to cover numerical correlation tools. These too have a much deeper and richer history than is evident in the article, and the portrayal is not entirely accurate about those connected to the GBDB. In very different ways, SinoCor and CONOP built on two-dimensional graphical correlation, which was published by Alan Shaw in 1964. Like GraphCor before it, SinoCor is computer-assisted graphical correlation. Given the enormous hardware advances, SinoCor was able to present users with a more elegantly straightforward graphical user interface, which helps students understand graphic correlation. CONOP, by contrast, is multidimensional. It takes Alan Shaw's principle of "Economy-of-Fit" to a wider array of stratigraphic data types and separates the consideration of sequencing from spacing. In this regard, CONOP resembles Lucy Edwards' "No-Space Graphs" from the 1970's. Another important biostratigraphic sequencing tool, Biograph, supports important macro-evolutionary studies, but is not mentioned. I would hope to hear of plans to incorporate time series tools that are well developed for the correlation of geochemical data and in astrochronology. For greater efficiency in sequencing huge datasets, which might be called big-data among paleobiologists, the original GBDB developers have since modified CONOP to CONOPSAGA. That development supports the most recent of the papers cited here and on the GBDB website, but the enabling software development is not mentioned.

Two other numerical sequencing tools that appeared years ago deserve mention: Jean Guex' Unitary Associations (published in book form as early as 1991, and the basis of Biograph) and John Alroy's CONJUNCT (also Appearance Event Ordination). They are especially relevant because they rely upon co-occurrences of taxa, the kind of information that is readily extracted from the collection-based PBDB. Guex and Alroy have separately made the case that co-occurrence is more trustworthy stratigraphic information than the order of all first- and last-occurrences in a single stratigraphic section.

These are arguments for including collections that lack the context of a stratigraphic section together with the section-based data. A collection is a stratigraphic section of restricted extent; the limiting case, if you will. If the collection has been assembled across many beds, however, it is no longer evidence of coexistence – a problem more easily avoided or recognized in section-based data. It is unfortunate to call these collections "virtual" data. It is very odd to call borehole records "virtual" sections (line 92). The term "virtual" is better reserved for composite sections that have been built by combining data from multiple locations, as in graphic correlation. For these, the familiar image of a stratigraphic section serves to hold all compiled information about sequence and spacing, but unlike cores, has no physical existence.

The authors (line 58) imply that SinoCor has few users because its file format is unique, and yet their own GBDB is correctly claimed (line 55-56) to readily export data for the software tool. Perhaps the authors overestimate the market for any of these tools. No matter, tool development has moved on. The GBDB needs to consider new developments like CONOPSAGA, Dynamic Programming (with its portfolios of alternative hiatus treatments), AstroChron and its potential hybrid AstroConop.

The authors' remarks about fossil terrestrial organisms are surprising. They recognize John Alroy as a key contributor to PBDB-based publications on biodiversity. John's much earlier dissertation and the publications that established his expertise were exhaustive big-data(?) analyses of all Cenozoic land mammal fossils of North America. John is another product of the Chicago school; he writes paleontological database management scripts and sequencing tools that have been integral to the PBDB. Vertebrate paleontologists might be disappointed to learn that the GBDB does not yet extend to collections younger than Eocene (line 162). Most professional paleontologists will know that section-based data are inherently more sparse for non-marine than marine organisms.

The article repeatedly emphasizes the section-based structure of the GBDB, and yet risks undermining this essential quality. The authors seem to be critical that "for a long

time the GBDB focused on stratigraphic records instead of only fossils" (lines 169-171). And yet the leaders of the project were all paleontologists. The impressive record of their publications from the GBDB, as evident from this article and the GBDB web site, is overwhelmingly paleontological. The key to better resolving power in the geologic history of taxon richness is unambiguous evidence of superposition and sequence. That evidence is more richly contained in sections with multiple, ordered collections of fossils, than in individual collections. Lithologic records hold clues to hiatuses and habitat biases. This is the one of the distinguishing advantages that the GBDB gave to paleobiologists.

The authors and GBDB managers need to distinguish those uses that are essentially paleobiologic from those that are biostratigraphic. There are studies that legitimately build from collections. There are those that are concerned with high-resolution sequencing. The latter are the vital constituency of the GBDB and it is these publications that will drive success. Some of these potential customers build their own databases and manage their own quality control, not relying on either the GBDB or the PBDB except as an aid in finding publications. The PBDB began that way. Paleozoic ammonoid experts, for example, have built their own database (GONIAT.ORG), which some find more open and user-friendly than either the GBDB or the PBDB. For Mesozoic and Cenozoic micro-paleontologists, there is MICROTAX.ORG. Other open paleontologic databases cover Baltic cores, Ocean drilling data and Cretaceous sections, for example. These databases and others like them have the advantage of narrower scope than the GBDB and PBDB. They set a high standard for completeness, expert authentication and community involvement

The article provides two numbered lists: the first presents customer dissatisfaction with the GBDB (no statistics, dates or sources given); and the second presents implemented and planned improvements. Neither list is particularly specific. It would place the GBDB in more positive light to emphasize the upgrades already in place and to show that these are responses to customer wishes. Some of the upgrades concern

flashy look-and-feel. The serious potential users, whose papers draw envious attention to the GBDB, want better control of downloads, versatile analytical tools (downloadable and on-line web services), and an assurance of quality control. It is also reassuring to see that the managers of a database trust its quality for their own high-profile, collaborative publications. Quality control has two levels: 1) accurate transcription of published information; and 2) expert cleaning and updating of legacy information. The PBDB, for example, seems to promise users an updating of the higher taxonomic assignment of species and a modern evaluation of synonymy. The breadth and expertise of the data compilers and authenticators may impress users more than the number of data transcribers who may not be accomplished taxonomists. The GBDB opinions are potentially an attractive feature. In the past, some users found that they could know that there was an opinion, but had difficulty accessing the source or the content of the opinion. It is not clear whether this has changed. Expert cleaning can be more than perfect matching to the published source. Those who mine data from this primary literature notice some obvious mistakes in measurements, numbers and names; some are straightforward to correct; so too are some types of obsolescence.

"Big data" is a popular jargon phrase. Paleontologists have, indeed, been compiling large datasets for the past 2-3 decades. These are small data, however, compared with those that come from geoscientific instruments that continuously monitor seismic-waves, GPS satellites, weather stations or tide gauges, for example. By comparison, it seems to be an overstatement to use the term "big data" for a data set whose essence that can be downloaded to a single Excel flat file and readily manipulated on an old Core i5 laptop to select subsets of data and sources or generate a world map of localities. Presumably the GBDB itself is a true relational database with a far more versatile data structure.

The article concludes with a relatively weak return to what has been true in the past and to statements that are too generalized. That dissipates the excitement of reading that the GBDB is alive again. The three databases mentioned (there are others,

of course) still complement one another and still aim to be readily accessible. No-body should doubt that. Readers need more specific information about the updates and reassurance that the database is still managed by experienced geobiologists and stratigraphers. The rate at which GBDB employees can enter data is very impressive. One advantage of the PBDB, in the minds of many users however, is that data entry is a community effort involving authorized users, each with proven and vetted exper-tise. Published data include errors and obsolescence. Many of us trust the PBDB authorizers to improve the quality of the information during data entry.

One of the most difficult aspects of published paleobiological data is to continually im-prove upon the initial estimates of the numerical age of fossil occurrences. The GBDB offered an advantage over the PBDB, because the ease of recovering superpositional order of events from sections enabled users to recalibrate their composite data sets with the most recent geochronometry. I trust that the GBDB will continue to build on this research advantage.

Paleobiological databases of the future ought to be able to continually sequence global and regional virtual sections. When new data are entered, the database ought to be able to respond immediately whenever the new data would be the oldest or youngest find of a taxon or a previously unproven co-existence; that is, data that potentially change the global range of a taxon should be flagged as soon as possible after entry. That is just one example of a deeper intellectual advantage of summary graphics for the whole database. In other words, automated and continual sequencing of the fossil record could become part of the validation and cleaning stage in the GBDB schema. CONOP- or SinoCor-generated graphics of richness and extinction through time should become a routine splash-page of the database, just like paleogeographic summaries. Like the maps of the spatial distribution of data, these time-series graphics could indi-cate how data-support and uncertainty (rarefaction, perhaps) vary through time. Both could be incentives to contribute new data where coverage is weak.

DATA UNIQUENESS: Although all these data have been previously published in numerous papers, the uniqueness of the PBDB database is that so much information is all compiled in one place, and in one format. Most of the primary sources are in Chinese. As compiled here, the data are uniquely accessible to a much wider audience of potential users. The majority of the data were compiled and available through the GBDB years before this article was written. The article compares the total holdings of the GBDB and PBDB and contrasts the geographic distribution of their primary sources. From this we may assume that the overlap is not substantial, but there is no systematic analysis of duplication between the databases - another possible approach to quality control.

DATA USEFULNESS: The data have already proven their usefulness in several major publications about macro-evolution and mass extinction. Their usefulness is primarily via the tools of the GBDB. The Excel file made available here might improve the usefulness for some researchers, but the data management tools of the GBDB are almost certainly superior.

DATA COMPLETENESS: The data are surely the most comprehensive of their kind for localities in China. The entire data set has been posted in the flat Excel file.

DATA QUALITY: The GBDB makes a good faith effort to ensure that the data are true to the original publications. Corrections for error and obsolescence are handled by "opinions." There is no guarantee that the opinion process amounts to a complete assurance of quality. The best assurance of the quality of these data is that they are not only already included in a permanent and managed repository, but they have also been used in major research articles that have stood the test of peer review and publication.

PRESENTATION QUALITY: The article is an abbreviated retrospective on the history of the GBDB. It is less about the data themselves than an advertisement for on-going and future changes in the GBDB platform that makes these data public. There is nothing in the article that would be at odds with the data themselves, but there are statements that might undermine readers' confidence that the database management team has a

sufficient breadth of expertise. Perhaps the need for a brief article has produced a condensed history that readers may find deficient and not always accurate. This problem for readers' confidence in the future of the GBDB is compounded by a tendency toward general rather than specific wording and all the usual misunderstandings that arise in translation between the Chinese and English languages. These linguistic flaws would be relatively easy for a native English-speaker to correct.

ACCESSING THE DATA: I navigated to the Xenodo site from the PDF abstract and was able to download the GBDB "all section" data as a Microsoft Excel flat-file with 764545 rows. Each row appears to be one taxon assigned to a collection with lithological information. From this file I easily generated my own world map of the data locations, using Excel functions. The location symbols produce a recognizable world map, as expected, with dense coverage for China, sparse patterns recognizable as Europe and North America.

The abstract posted at the Xenodo site provides a web address for the GBDB. [https://www.geobiodiversity.com/] This address could not be reached in several attempts; it finally responded slowly with the new statistical splash page for the GBDB. The abstract submitted for review does not end with this url.

The new website presents users with some problems and disappointments. The "fossil ontology" label is intriguing, but clicking it does not work. Not all buttons in the bar at the bottom left do anything noticeable and not all have a call-out explanation. The paleogeographic maps that summarize section locations are fascinating to play with via the circular geologic time scale. The popup words to explain the eye icons are covered by the cursor and cannot be read. Occurrences is mistyped as "Occureences." The publications have urls, but these are not clickable. The site is clearly new, more impressive than the old one in some ways, but still not fully fledged.

Google-searching for the GBDB on line yielded only the old Geobiodiversity Database site "Powered by Junxuan Fan Copyright 2006-2017 Released version: 1001130." The

newest posting on that site is from January 2018. Manuscript first-author Xu Honghe is listed as the contact. Perhaps that site should help potential users by linking them tothe new site. The new site does include a link to the old; this may be welcomed by some long-time users.

---

## Referee Comment (RC2) · Richard Butler (Referee) · 20 Sep 2020

GENERAL COMMENTS This article has the potential to provide a useful and welcome introduction to the history, structure, content, functionality and analytical tools, and proposed future of the Geobiodiversity Database, a database with huge research potential that remains much less well known and used within the international geoscience community than the US-based Paleobiology Database. The current version of the manuscript does not, however, fully achieve its goals. It is lacking in details in some parts, particularly those dealing with historical aspects, and at times difficult to follow and somewhat repetitive. In this review I first provide some major comments on

three particular sections of the manuscript, and then more detailed comments on other issues with the text.

The history section (lines 35–80) is rather opaque and should provide more details. It would be useful to know (lines 36–45), at which institutions, by which researchers, and using what sources of funding the GBDB was established, how the database was managed, where the data enterers were based etc. In lines 38–39 a "large palaeobiology database" is mentioned, but it is not made explicitly clear that the authors are referring to (I believe) the US-based Paleobiology Database. I am not sure that the statement that the PBDB "temporarily ignored" Chinese data is correct: such data may have been underrepresented, but the PBDB currently includes >13,000 Chinese collections, and several thousand of these were entered prior to 2007. On line 44 there is mention of aligning data entry with standards of "international researchers", but it is not clear what those standards are. On lines 51–54 a number of different statistical and visualisation tools are mentioned by name, but these should be explained in more detail. On lines 56–59, stratigraphic correlation tools (CONOP, SinoCor) are mentioned, but not explained with sufficient detail for readers who are not already familiar with them.

In the section on the data of the GBDB, some aspects of the structure of the database are described with insufficient detail to be understandable. For example:

- Lines 90–91: it is not clear what a 'virtual section' is. This needs expanding and explaining with further detail. It is unclear to me why a fossil without any detailed associated stratigraphic section and a borehole (which presumably has a detailed record of changes in sedimentology) both represent 'virtual sections' as they seem like quite fundamentally different kinds of data.

- Lines 95–96. It is very unclear how the palaeontological data in the GBDB are linked to sections, and how this relates to occurrence-based datasets. Some of the statements made about this seem internally inconsistent.

- Lines 100–101. It is unclear what "opinion data" are here, and how taxonomic opinions are treated by the GBDB. Are these taxonomic opinions, and reflect changes in the identification of fossils from particular sections? Does the change in the taxonomic identity of a fossil from a particular section reflected in other sections at all? Is there an overarching taxonomic framework, similar to the dynamic taxonomy present in the PBDB?

- Do the GSSPs included in the GBDB only include those from China, or is this global?

- Lines 105–107. This collaboration with BGS is very incompletely described. Is data compiled in the same way as in the rest of the database? Is the data accessible to other researchers?

- Lines 108–109. Are these borehole data from oil companies publicly accessible, despite potential commercial sensitivities? Should be made explicit.

The section on newly-added data in the GBDB contains comments that are questionable in parts. For example, although it is undoubtedly true that marine invertebrates have long been the core focus of Phanerozoic diversity studies, it is not true to imply that there have been almost no studies of terrestrial diversity, as is done here where only two local studies of Chinese Palaeozoic diversity are cited. For example, considerable work has been conducted on global land plant diversity, starting with the work of Andrew Knoll, Karl Niklas and Bruce Tiffney in the late 1970s through early 1990s, and continued today by other researchers such as Borja Cascales‐Minana. There is a long track record of studies of terrestrial tetrapod diversity, beginning with the work of Mike Benton on global patterns numerous studies by John Alroy and others on Cenozoic mammal diversity, and then a huge number of papers over the last 15 years on diversity patterns in individual clades, from dinosaurs to hominids. Many of these have used PBDB data and are listed on the official publication list of the database. Finally, there is also a long history of studies of global insect diversity, going back nearly 30 years. This section should be revised in light of this extensive history of terrestrial research.

SPECIFIC COMMENTS The title of the paper is somewhat unclear in its meaning, and I would suggest changing it for something simpler and more accessible. Perhaps something like "The past and future of the Geobiodiversity Database: a section-based stratigraphic and palaeontological database"

There is much use of the term "big data". But the size of the datasets contained within the Geobiodiversity Database and other comparable databases (PBDB etc.)  would generally not qualify as big data under most definitions. A slightly more neutral term, such as "large data sets" might be more appropriate.

Line 12: should be "Here, a thorough introduction is given to the Geobiodiversity Database"

Line 13: in various places you use the word "serial" (here "serial of scientific studies") when "series" would be correct

Line 14: "Nevertheless, the existing problems of the GBDB limited the using of its data". This phrasing is problematic in the abstract because the "existing problems" have not been described.  I would suggest combining this and the following sentence into something like "Nevertheless, limited use of the GBDB by the wider palaeontological community led to reorganisation and improvements beginning in 2019".

Line 18: "Further collaborations are proposed" – this is not really developed in the paper, and it would be good to know if, for example, more definite discussions on collaboration have been had with other database leadership teams.

Lines 19–20: This statement on the availability of datasets – should it be in the abstract?

Lines 22–25: These opening lines are quite repetitive. Would suggest rewording to reduce repetition e.g. "Palaeontology and stratigraphy have become increasingly quantitative branches of geoscience in recent decades (REFERENCES). Quantitative analyses of large datasets of fossil and stratum records have become more common in

studies of. . .

Lines 25–27: rather a brief selection of papers are cited as evidence of the increasingly quantitative nature of palaeontological/stratigraphic research, and they are fairly biased towards studies linked to the GBDB. A broader range of citations would be useful here.

Line 27: I would suggest "academic databases", rather than "professional databases"

Line 30: I would suggest using the term "user-friendly" rather than "friendly" when talking about the accessibility of the database for users.

Line 55: would suggest "unique" rather than "exclusive"

Line 158: opinions in the PBDB are taxonomic opinions, not palaeobiological opinions.

Line 170: the relationship of taxon occurrences to sections needs to be explained in more detail in this manuscript. Do taxa occur associated with distinct horizons within a section (in which case this would quite closely approximate an occurrence-based dataset)? Or are occurrences clumped within sections in some way?

Lines 193–194: It is unclear what is meant here. Do you mean that the GBDB was not being backed up and that its use was hazardous because of the potential for data loss? Needs some more clarity.

Lines 195–197: More information is needed on what a "safe data bank" is.

Lines 198–200: The description of the data entry process is unclear. Who are registered authorizers, and how are they selected? Can anyone enter data, but it has to be checked by a registered authorizer?

---

## Author Comment (AC1) · 4 Oct 2020

Corresponded author's respond to the first reviewer's comment

Thank Peter for kind and nice comment and revision suggestions of the manuscript. We revised the manuscript carefully. The reviewer gave detailed information about the palaeobiological database and the history. We really benefit.

The manuscript is not a review but a description of the data the GBDB has and we also introduce something new of the database and the website, something we did after the change of the managements of the GBDB. The turnover occurred at the end of 2018

and my taking over occurred in the middle of 2019. There are too many things beyond the academic. But the palaeontology community is so small, even our colleagues in the Europe and the USA seem to know and spread gossips about the database. This is not good. I admit that in the first version of the manuscript there are something negative and critical. But in the new version, I deleted them. I see the disadvantages of the GBDB, make the updates, and look ahead.

After the turnover of the GBDB a survey is given in the community. we received some feedbacks and found the existing problems of the GBDB. Such represent the users wishes of the GBDB that is why we made the updates. We are confident to the future of the GBDB and the thing we are doing.

I am sorry to say that this manuscript is not a review of the quantitative study of palaeontology and stratigraphy, or a review of the palaeontological database. Deep knowledge of the numerical study of the palaeontology and stratigraphy is not given here. and we don't think it is proper here. Previous papers published by Fan et al gave these introductions. We here only give the retrospect. Previous papers stated little on the data structures or comparisons. We don't do the duplicate job.

The reviewer's suggestion on the correlation software, such are CONOP and SinoCor, is followed. Accordingly, the manuscript is revised.

About the terrestrial organism fossil records, we did more checking and realized the importance of J Alroy' work. We here want to emphasize that the terrestrial fossil records had been neglected for a long time till we collect these data into the GBDB. These parts were revised in the new manuscript accordingly.

About the section records of the GBDB. We realized the previous downloading result shows only section result and is not compatible to the fossil occurrences result. Then we updated the database, making the flexible searching and downloading options. Users can download either data individually.
About the big data. we admit the data volume of ours is relatively not big enough than other fields. We here just emphasized the change of the study method and are hoping to promote the data-driven study. Actually, using data of GBDB several had output several impressive scientific results.

The data uploaded to the Xenodo include the section data of the GBDB, one can use them to do the palaeogeographical and biodiversity or other related study, just as previous authors did. The website sometime is not quick enough, probably because that the sever is in China and the too many data are loading when the first time visiting. The fossil-ontology represents our next plan for the database which is for the non-structured data of the fossils and strata.

---

## Author Comment (AC2) · 4 Oct 2020

Thank Richard's comment.

About the history of the GBDB data. As we mentioned previously this manuscript is not a review of databases or quantitative study methods of palaeontology and stratigraphy. It is only a description of data that GBDB has and is having, avoiding the duplicate parts in the previous publications. The PBDB's work in emphasized in the revised manuscript and the related introduction is changed accordingly.

The 'virtual section' is not used any more. Here we want to state that some fossil

collections were treated as a from a section. The explanation is given in the revised manuscript.

About the data structure. GBDB is section based and its data are compatible to fossil occurrences. In our updates, one can search and choose the data format and data result. This part is given more details and explanations.

There is more information about the opinion data in the revised manuscript.

GSSP had been thought to be included in the database as both records (existing data) and panorama images, but currently, only a few GSSP is included. The related work is still awaiting.

Only a bit of BGS data are accessible to researchers, for the sake of the agreement of the BGS and the GBDB. The same things occur about the data from the oil company.

There is much work about the fossil records of the terrestrial organism. The section is revised and much work is mentioned. Thank the two reviewers.

The title was change to 'Retrospective and prospects of the GBDB. . ..' According to reviewers' comments. It is not a review, but a data description and introduction.

About the big data. we admit the data volume of ours is relatively not big enough than other fields. We here just emphasized the change of the study method and are hoping to promote the data-driven study. Actually, using data of GBDB several had output several impressive scientific results.

Specific revisions are given in the revised manuscript.

———————————————

---

## Author Comment (AC3) · 4 Oct 2020

According to the reviewers' comments and suggestions, the corresponded author made the revised manuscript, which can be upload now. The revision mainly includes the following aspects.

The negative and unnecessary criticisms were deleted from the manuscript. The updates of the GBDB and the website focused on the survey and users wishes. The positive way is looking ahead. The is need to the development of the GBDB.

Specific statement and explanations are given to some concepts and unclear points,

making the data description clear.

Some important work was introduced for the understanding of the data and the study, the references are added.

Some spelling and grammar mistakes were removed

––––––––––––––––––––––––––––––

---

## Referee Report (RR1)

The revisions to this article strengthen its effectiveness as a bearer of welcome news for analysts of macroevolution: the GeoBiodiversity DataBase (GBDB) is moving forward again after it appeared to lose impetus during a management turnover. More a prospectus or status report than a research report, the article serves the purpose of alerting potential users to updates planned for the database and its web portal. These plans are now more clearly connected to market research – a survey of users. The GBDB does not need a separate data-description paper; this is a database-description paper. As before, I assume, the database can be downloaded as an Excel flat file. The flat file is surely less versatile than the GBDB. The new portal to the GBDB website is already accessible and most of its features are functional.

The revisions are a good faith response to the on-line reviews. The authors have eliminated some tangential matters and clarified terms that had puzzled reviewers. Their revised manuscript focuses more effectively on the current database and softens implied criticisms of its initial design. The reference list is now a richer account of previous analyses of macroevolution that the GBDB data had made possible

The manuscript might seem a little long for its rather straightforward message, but the retrospective parts could be of value to readers who are not yet familiar with the GBDB and the way its structure and geographic scope complement the earlier established PaleoBiology DataBase (PBDB). This reviewer had had attended workshops with the founders of the GBDB. Many potential readers may not have had this advantage.

If the editorial decision is to publish the manuscript, I trust there is sufficient staff support for thorough copy editing. There remain those almost inevitable grammatical glitches that arise between the Chinese and English languages. In most cases the meaning is not lost; the glitches are just a distraction. In a few cases the author's intent might be obscured, at the sentence level, but not for the larger status report. I would prefer "status report" in the title instead of "retrospects and prospects," which seems at odds with the authors' response that "this manuscript is not a review." This is a small point, however.

---

## Author Response (AR3)

According to the reviewers' comments and suggestions, the corresponded author (Hong-He Xu, hhxu@nigpas.ac.cn) made the revised manuscript, which can be upload now. The revision includes the following aspects.

5 "retrospect and prospects" in the title was changed. Now the title is "A status report on a….." which seems better for the whole manuscript.
The grammar and spelling of the whole text are checked carefully to avoid errors as possible as we can. The British spelling is used, e.g. we use "palaeo-' instead of "paleo-", except some special terms.
The data volumes of the compared database were updated to the November of 2020.

[revised manuscript text omitted]

*The stratigraphic formation data of the PBDB was obtained from Prof. W. Kiessling whilst one can see these records from the portal of the PBDB.